# Repurposing the selective estrogen receptor modulator *bazedoxifene* to suppress gastrointestinal cancer growth

Pathum Thilakasiri[1], Jennifer Huynh[1], Ashleigh R Poh[1], Chin Wee Tan[2], Tracy L Nero[3,4], Kelly Tran[1], Adam C Parslow[1,5], Shoukat Afshar-Sterle[1], David Baloyan[1], Natalie J Hannan[6], Michael Buchert[1], Andrew Mark Scott[1,5,7], Michael DW Griffin[4], Frederic Hollande[8], Michael W Parker[3,4], Tracy L Putoczki[2], Matthias Ernst[1,*,†] (iD) & Ashwini L Chand[1,**,†] (iD)

## Abstract

Excessive signaling through gp130, the shared receptor for the interleukin (IL)6 family of cytokines, is a common hallmark in solid malignancies and promotes their progression. Here, we established the *in vivo* utility of *bazedoxifene*, a steroid analog clinically approved for the treatment of osteoporosis, to suppress gp130-dependent tumor growth of the gastrointestinal epithelium. *Bazedoxifene* administration reduced gastric tumor burden in $gp130^{Y757F}$ mice, where tumors arise exclusively through excessive gp130/STAT3 signaling in response to the IL6 family cytokine IL11. Likewise, in mouse models of sporadic colon and intestinal cancers, which arise from oncogenic mutations in the tumor suppressor gene *Apc* and the associated β-catenin/canonical WNT pathway, *bazedoxifene* treatment reduces tumor burden. Consistent with the proposed orthogonal tumor-promoting activity of IL11-dependent gp130/STAT3 signaling, tumors of *bazedoxifene*-treated *Apc*-mutant mice retain excessive nuclear accumulation of β-catenin and aberrant WNT pathway activation. Likewise, *bazedoxifene* treatment of human colon cancer cells harboring mutant *APC* did not reduce aberrant canonical WNT signaling, but suppressed IL11-dependent STAT3 signaling. Our findings provide compelling proof of concept to support the repurposing of *bazedoxifene* for the treatment of gastrointestinal cancers in which IL11 plays a tumor-promoting role.

**Keywords** colon cancer; gastric cancer; gp130; interleukin-11; interleukin-6
**Subject Categories** Cancer; Digestive System; Pharmacology & Drug Discovery

## Introduction

The contribution of inflammatory cytokines to the progression and treatment resistance of solid cancers is now widely accepted, even for malignancies that occur in the absence of overt inflammation (Putoczki *et al*, 2013). Among the cytokines most prominently involved in this process are those of the interleukin (IL)6/11 family, characterized by the shared use of the transmembrane receptor β-subunit gp130 (Ernst & Putoczki, 2012; Putoczki *et al*, 2013). We have recently identified a hitherto unrecognized tumor-promoting role for IL11 in mouse models of gastrointestinal cancers that arise in the mucosal epithelium of the stomach, small intestine, or colon (Ernst *et al*, 2008; Putoczki *et al*, 2013). Strikingly, genetic restriction of IL11 signaling by either ablation of the IL11Rα co-receptor subunit, or therapeutic administration of the antagonistic "IL11-Mutein" variant, limits the growth of tumors driven by excessive signaling of the canonical WNT pathway resulting from oncogenic β-catenin/*CTBNN1* or *APC* mutations that underpin 80% of human colon cancer (Putoczki *et al*, 2013). Meanwhile, others have proposed a role for IL11 signaling to enable metastatic spread and to retain tumor heterogeneity that underpins an aggressive phenotype (Kang *et al*, 2003, 2005; Marusyk *et al*, 2014). We therefore proposed that tumors exploit the IL6/11 family cytokines, which

1 Olivia Newton-John Cancer Research Institute, School of Cancer Medicine, La Trobe University, Heidelberg, Vic., Australia
2 The Walter and Eliza Hall Institute, Melbourne, Vic., Australia
3 ACRF Rational Drug Discovery Centre, St Vincent's Institute, Melbourne, Vic., Australia
4 Department of Biochemistry and Molecular Biology, Bio21 Institute, University of Melbourne, Melbourne, Vic., Australia
5 Department of Molecular Imaging and Therapy, Austin Health, Melbourne, Vic., Australia
6 Department of Obstetrics and Gynaecology, University of Melbourne, Melbourne, Vic., Australia
7 Department of Medicine, University of Melbourne, Melbourne, Vic., Australia
8 Department of Clinical Pathology, University of Melbourne Centre for Cancer Research, Victorian Comprehensive Cancer Centre, University of Melbourne, Melbourne, Vic., Australia
  *Corresponding author. Tel: +61 3 9496 9775; E-mail: matthias.ernst@onjcri.org.au
  **Corresponding author. Tel: +61 3 9496 9373; E-mail: ashwini.chand@onjcri.org.au
  †These authors contributed equally to this work

provide a rheostat function to link an effective, proliferative wound-healing response to the ensuing inflammatory response of the intestinal epithelium (Ernst *et al*, 2014). We surmised that the activity of these cytokines becomes rate limiting for the growth of tumors and can be exploited as therapeutic vulnerability.

Due to their solubility and cellular expression, cytokines and their receptors are preferred targets for the development of antibody-based therapeutics and have become mainstream targets for the treatment of inflammation and other chronic diseases. Accordingly, antibodies directed against IL6, or its cognate ligand-specific IL6Rα co-receptor subunit, are either already in the clinic, or in advanced clinical trials for the treatment of autoimmune conditions and some hematological malignancies. Surprisingly, no such clinically approved antibody reagents have been developed to target IL11 signaling either at the level of the ligand or the IL11Rα receptor subunit.

Previous *in silico* modeling has revealed that the selective estrogen receptor modulators (SERM) *raloxifene* and *bazedoxifene* could interfere with the protein–protein interactions between IL6 and gp130 (Li *et al*, 2014). These third-generation SERMs are approved in Europe and the United States for the prevention and treatment of postmenopausal osteoporosis without conferring agonist effects on endometrial, ovarian, and breast tissues (Silverman *et al*, 2008; Pinkerton *et al*, 2012; Flannery *et al*, 2016). Here, we provide the first *in vivo* evidence that *bazedoxifene* treatment of mice, which harbor epithelial tumors in the glandular stomach, the small intestine or the colon, with drug doses corresponding to treatment regimens for osteoporosis patients, suppresses tumor growth irrespective of the gender of the host. Akin to our observations with the IL11Rα receptor antagonist IL11-Mutein (Putoczki *et al*, 2013), we find that *bazedoxifene* restricts the growth of intestinal tumors by suppressing IL11-mediated signaling rather than by interfering with excessive canonical WNT signaling that arises from bi-allelic inactivation of the *Apc* tumor suppressor gene. Collectively our observations suggest that *bazedoxifene* could be readily repurposed for the treatment of gastric and colon cancers and that *bazedoxifene* serves as a tool compound for further chemical refinements to increase specificity and affinity of future small molecule IL11 signaling antagonists.

## Results

### *Bazedoxifene* blocks IL11 signaling

*Bazedoxifene* is thought to inhibit IL6 signaling by interfering with the formation of the signaling-competent hexameric receptor complex. *Bazedoxifene* prevents the aggregation of two trimeric receptor complexes, comprised of an IL6 ligand, an IL6Rα co-receptor subunit, and one gp130 subunit, and resulted in suppressed activation of STAT3 (Li *et al*, 2014). Given the proposed similar hexameric 2:2:2 nature of the IL11:IL11Rα:gp130 complex (Veverka *et al*, 2012; Putoczki *et al*, 2014), we investigated whether *bazedoxifene* could also inhibit IL11-mediated, gp130-dependent STAT3 signaling. We co-expressed human IL11Rα alongside the STAT3-responsive pAPRE-luciferase (luc) reporter construct in HEK293 cells. Treatment with IL11 induced a 15-fold increase in APRE-luc

reporter activity, which was antagonized in a dose-dependent manner by *bazedoxifene* (Fig 1A and Appendix Fig S1A). To ensure this was not a generic effect conferred by antagonistic-acting estrogen analogs, we also tested these cells with *tamoxifen* (Appendix Fig S1A). *Tamoxifen* failed to suppress IL11, suggesting a selective effect of *bazedoxifene* in the inhibition of IL11:IL11Rα:gp130 signaling.

To extend our findings to a more biologically complex response, we exploited the IL3-dependency of the murine BAF/03 pro-B-cell line for their survival and proliferation *in vitro* (Hilton *et al*, 1994; Nandurkar *et al*, 1996). For this, we installed expression constructs encoding gp130 alongside either IL6Rα, IL11Rα, or LIFR in BAF/03 cells. This enabled the subsequent propagation of the corresponding BAF/03 clones with IL6, IL11, or LIF, respectively, in the absence of IL3 (Fig 1B and C). We confirmed that *bazedoxifene* treatment antagonized IL11-mediated cell proliferation in a concentration-dependent manner (Fig 1B). Corroborating the selective effect that we observed with *bazedoxifene* on STAT3 transcriptional activity in HEK293T cells, we also found that *bazedoxifene* but not *tamoxifen* suppressed IL11-mediated proliferation of BAF/03 cells expressing human IL11Rα (Appendix Fig S1B). Our observations are consistent with the proposed inhibitory mechanism of *bazedoxifene* on the hexameric gp130 signaling complex, as *bazedoxifene* also inhibited IL6-dependent BAF/03 proliferation (Fig 1B). By contrast, *bazedoxifene* treatment not only failed to antagonize IL3-dependent parental BAF/03 cell proliferation, but also that of the LIFR-expressing clones stimulated with human LIF (Fig 1C) and consistent with LIF forming trimeric LIF:LIFR:gp130 complexes (Gearing *et al*, 1991; Hammacher *et al*, 1998). We also excluded that *bazedoxifene* interfered with gp130 signaling in the absence of ligand or of receptor α-subunits. For this, we exploited a synthetic form of gp130 in which a leucine zipper region of c-jun substitutes for the native extracellular receptor domain and confers ligand-independent homodimerization of the resulting chimeric L-gp130 proteins (Stuhlmann-Laeisz *et al*, 2006). Thus, L-gp130-dependent BAF/03 cell proliferation should be refractory to *bazedoxifene* inhibition, which we confirmed experimentally (Fig 1C). We surmise from this collective functional data that *bazedoxifene* disrupts IL11 signaling akin to its proposed action on the signaling-competent, hexameric IL6 receptor complex.

### *In silico* modeling of *bazedoxifene* bound to gp130 Site III residues

It was been previously predicted that *bazedoxifene* competes with binding of IL6 in the trimeric IL6:IL6Rα:gp130 complex. It is the interaction between IL6 of one trimer and the gp130 co-receptor of the second trimeric unit that is inhibited thereby preventing the formation of the signaling-competent hexameric complex (Fig 2A and B; Li *et al*, 2014; Wu *et al*, 2016). Specifically, structural analysis of the IL6:IL6Rα:gp130 hexameric complex (PDB ID: 1P9M) revealed that IL6 binds to gp130 via residues leucine-57, glutamic acid-59, asparagine-60, leucine-62, tryptophan-157, and leucine-156 in Site III. A 30 Å resolution cryo-electron map of the IL11:IL11Rα:gp130 hexameric complex suggested that its subunits are organized in a similar arrangement to that of the hexameric IL6:IL6Rα:gp130 complex (Neddermann *et al*, 1996; Matadeen *et al*, 2007; Putoczki *et al*, 2014).

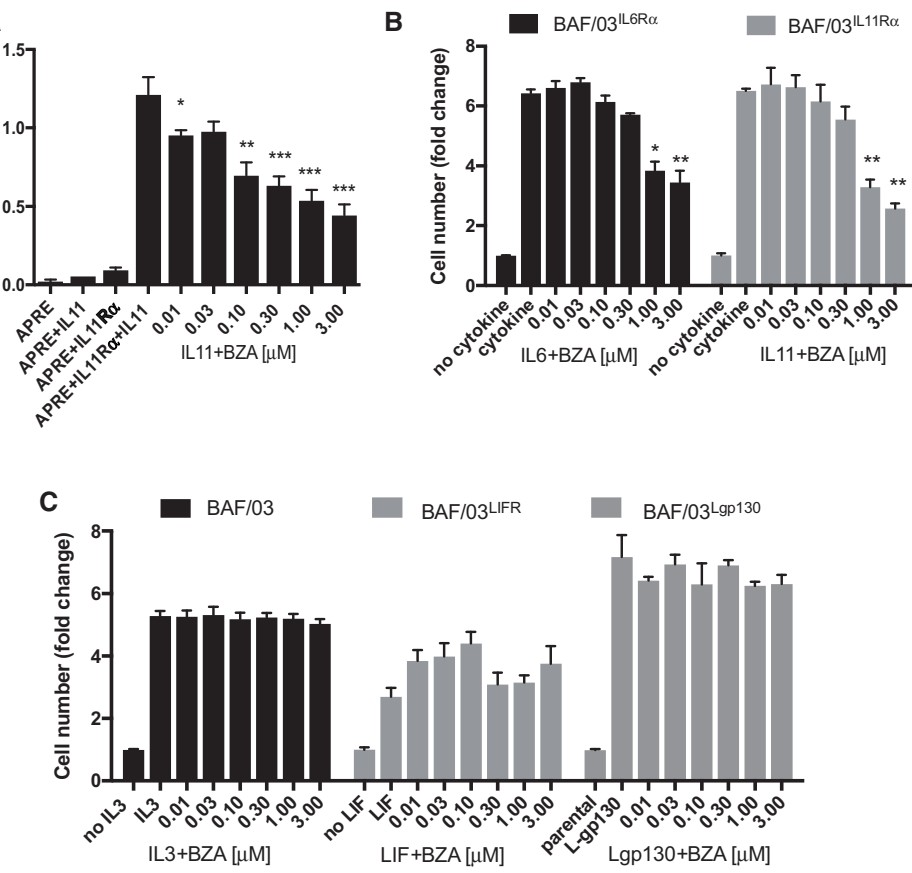

**Figure 1.** *Bazedoxifene* **suppresses IL11-mediated STAT3 signaling activity.**

A  Effect of *bazedoxifene* (BZA) on IL11-induced and STAT3-dependent pAPRE-firefly luciferase reporter activity in HEK293T cells expressing human IL11Rα. Cells were co-transfected with a non-responsive Renilla luciferase plasmid. Results are expressed as relative luciferase units (RLU), that is, firefly luciferase activity normalized against Renilla luciferase activity in each individual culture.

B  Effect of BZA treatment on proliferation of IL11 stimulated BAF/03 murine B-cell lines, as determined by MTS-assay. IL6 stimulation was used as a positive control. Cells were engineered to express human either IL6Rα or IL11Rα, respectively.

C  Effects of BZA treatment, as determined by MTS-assay, on parental BAF/03 cells stimulated with IL3, of LIF receptor (LIFR)-expressing cells stimulated with LIF, or of cells expressing the constitutive active L-gp130 construct.

Data information: Data are mean ± SEM, n = 3 individual cultures, *P < 0.05, **P < 0.01, ***P < 0.001, 2-way repeated-measures ANOVA, Tukey's multiple comparison test.

Site-directed mutagenesis studies indicated that IL11 interacts with gp130 at Site III via tryptophan-168 and IL6 interacts via tryptophan-157 as amino acids that form part of helix D of these ligands (Barton *et al*, 1999). We therefore aligned helix D from the crystal structure of IL11 (PDB ID: 4MHL; Putoczki *et al*, 2014) with that of IL6 to model how IL11 is likely to interact with gp130 at Site III (Fig 2C). When aligned in this manner, tryptophan-157 in IL6 and tryptophan-168 in IL11 indeed interact with the same gp130 residues (Fig 2D).

We next used our own unbiased *in silico* modeling approach and identified two possible binding modes of *bazedoxifene* with gp130 (Appendix Fig S2A–E). In the first one, the indole ring and azepanyl ring of *bazedoxifene* mimic the interactions between gp130 and the tryptophan-157 and leucine-57 residues in IL6 (Appendix Fig S2C and D). The indole ring hydroxyl substituent of *bazedoxifene* is able to interact through hydrogen bonds specifically with glutamine-78 in gp130, and the indole

and phenol rings both form π-π interactions with tyrosine-94. Meanwhile, the leucine-3 residue in gp130 engages through hydrophobic interactions with the indole ring and phenol substituent, as well as with the pendant phenyl ring of *bazedoxifene*. Finally, the *bazedoxifene* azepanyl ring extends in a largely hydrophobic region of gp130 comprising cysteine-6, cysteine-32, phenylalanine-36, isoleucine-83, and glutamine-91. In an alternative second binding model, the phenol ring and azepanyl ring of *bazedoxifene* mimic the interactions between gp130 and tryptophan-157 and leucine-57 in IL6 (Appendix Fig S2E and F). In this binding mode, the *bazedoxifene* indole ring can form π-π interactions with tyrosine-94 in gp130. Meanwhile, the phenol substituent of *bazedoxifene* can form a hydrogen bond with asparagine-92 in gp130, and the indole ring hydroxyl substituent could form hydrogen bonds with either the hydroxyl group of tyrosine-94 or the side chain of glutamic acid-12. The leucine-3 residue in gp130 would then form hydrophobic interactions with

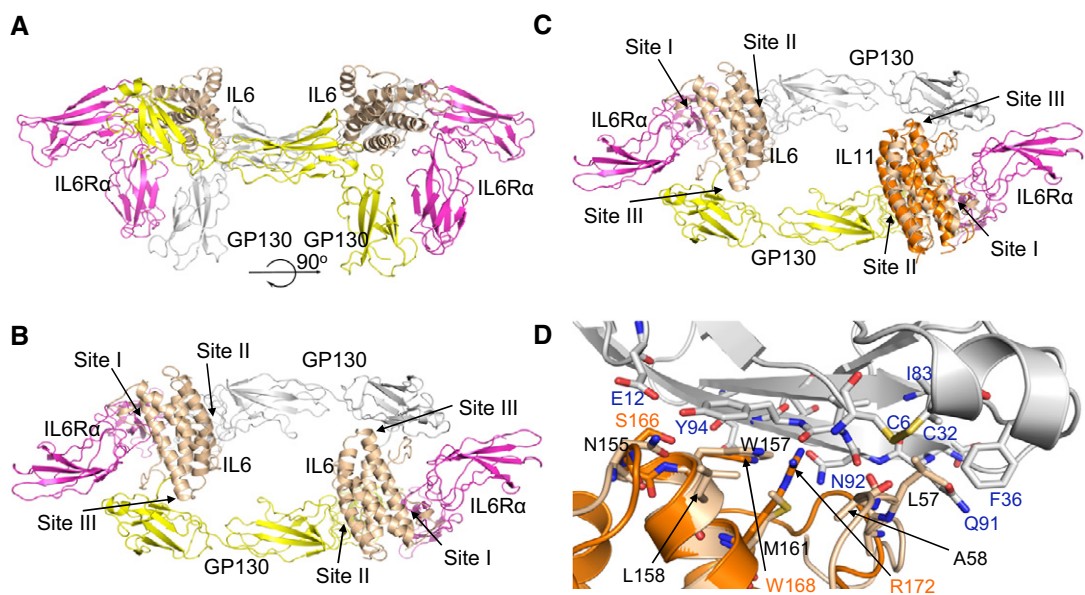

**Figure 2. Structure of the gp130/IL6 hexameric signaling complex and the corresponding gp130/IL11 complex.**

A   Crystal structure of the extracellular domains of the IL6 hexameric signaling complex (PDB ID: 1P9M; Boulanger *et al*, 2003), depicted in cartoon format. The signaling complex consists of two molecules each of IL6 (cream colored), IL6Rα (magenta), and gp130 (one colored gray and the second yellow).

B   90° rotation about the *x*-axis from the view shown in (A). IL6 binds with its specific IL6Rα subunit via interaction Site I and with gp130 via Site II and Site III. *Bazedoxifene* has been proposed to interact with Site III, thereby disrupting the interaction of IL6 and IL11 with gp130 (Li *et al*, 2014; Wu *et al*, 2016).

C   Proposed binding of IL11 to gp130 via Site III residues. The structure of IL11 (PDB ID: 4MHL, orange ribbon structure; Putoczki *et al*, 2014) was superimposed on IL6 in the hexameric receptor complex.

D   Close-up view of the Site III interaction sites between IL6 (cream ribbon, amino acid residues in black) or IL11 (orange ribbon, amino acid residues in orange) with gp130 (gray ribbon, amino acid residues in blue). The side chains of some of the residues at the Site III interface are shown as sticks.

the pendant phenyl substituent of *bazedoxifene*. Importantly, in either of the putative interaction models, *bazedoxifene* makes multiple interactions with Site III residues of gp130 consistent with the proposed models by Li *et al* (2014) and Wu *et al* (2016).

## *Bazedoxifene* inhibits IL11-dependent STAT3 activation and growth of patient-derived colon cancer organoids

Given our previous observations that IL11 signaling confers potent activation of STAT3 in gastrointestinal cancer cells (Putoczki *et al*, 2013), we next examined the effects of *bazedoxifene* on cytokine-stimulated human gastric and colon cancer cells. We observed that *bazedoxifene* treatment suppressed IL11-mediated induction of the transcriptionally active, phosphorylated STAT3 (pSTAT3) isoform in a dose-dependent manner (Fig 3A). Reduced pSTAT3 levels were also observed in the estrogen receptor (ER)-negative breast cancer cell line, MDA-MB-231, confirming the ER independence of the *bazedoxifene* effect on IL11/gp130 signaling (Fig 3A). To ascertain the biological importance of this observation in a pathologically relevant setting, we assessed the effects of *bazedoxifene* on IL11 signaling in patient-derived samples of colon cancer cells grown as primary cultures. We consistently observed that *bazedoxifene* treatment decreased pSTAT3 levels in a dose-dependent manner (Fig 3B). These tumor epithelial cells showed higher expression of *IL11* mRNA transcript when compared to that of *IL6* (Fig 3C). Furthermore, mRNA transcripts for *IL6R, IL11R,* and *SOCS3* were

readily detectable, with *IL6ST* transcripts, encoding gp130 as being the most abundant (Fig 3C). We then exploited patient-derived human colon cancer organoids grown in 3D cultures in the presence of IL11. We observed significantly smaller organoids in the presence of 10 μM *bazedoxifene* (Fig 3D and E). Collectively, our *in vitro* data in factor-dependent BAF/03 cells, human colon and gastric cancer cell lines, patient-derived colon cancer cells, and organoids provide compelling evidence that *bazedoxifene* effectively antagonizes IL11-elicited STAT3 signaling and associated cell proliferation *in vitro*.

## Therapeutic *bazedoxifene* treatment impairs IL11-dependent gastric tumor growth *in vivo*

In order to extend our *in vitro* findings to an *in vivo* cancer setting, we utilized the *gp130*[Y757F] mouse model of intestinal-type gastric cancer. In this model, gastric adenomas spontaneously and reproducibly develop with 100% penetrance in 4-week-old mice with an absolute genetic dependence on bi-allelic expression of the *il11rα* and *Stat3* genes, but completely independent of IL6 signaling (Jenkins *et al*, 2005; Ernst *et al*, 2008). Importantly, we had previously shown that therapeutic treatment of tumor-bearing, 10- to 13-week-old *gp130*[Y757F] mice with either the antagonistic IL11-Mutein peptide (Putoczki *et al*, 2013), JAK1/2 kinase inhibitors (Stuart *et al*, 2014), STAT3 antisense oligonucleotides (Ernst *et al*, 2008), or inducible short hairpin STAT3-RNA (Alorro *et al*, 2017), reduces tumor burden associated with reduced cell proliferation and increased apoptosis. Thus, we treated 13-week-old *gp130*[Y757F] mice with

**Figure 3. Bazedoxifene suppresses IL11-mediated STAT3 signaling activity in human cancer cell lines and patient-derived colon cancer primary cultures and organoids.**

A   Effects of *bazedoxifene* (BZA) on IL11-stimulated STAT3 activation (pSTAT3) in gastric (MKN1), colon (LIM2405), and breast (MDA-MB-231) cell lines.

B   Effects of *bazedoxifene* (BZA) on IL11-stimulated STAT3 activation (pSTAT3) in primary cell cultures of isolated colon cancer epithelial cells of three individual colorectal cancer (CRC) patients.

C   mRNA expression levels of *IL6, IL11, IL6R, IL11R, IL6ST* encoding gp130 and *SOC3*, a STAT3-regulated gene in epithelial cells derived from six colorectal cancer patients.

D   Effects of BZA treatment on IL11-dependent growth of human colon cancer organoids. Organoids were cultured for 6 days in the presence of IL11 (50 ng/ml) and then for a further 7 days in IL11 plus the indicated concentration of BZA. Representative bright-field microscopy images of organoid cultures at 7 days after BZA treatment. Scale bar = 300 μm.

E   Relative changes in organoid size after 7 day treatment with vehicle control or BZA as described in (D). Data are mean ± SEM, *n* = 3 individual cultures, one-way ANOVA, Dunnett's comparison test.

Source data are available online for this figure.

established gastric tumors for 7 weeks with *bazedoxifene* at a dose of 3 mg/kg i.p. five times per week (Fig 4A). This dosing was previously used to document inhibitory effects on estrogen-sensitive mouse tissue (Sakr *et al*, 2014; Flannery *et al*, 2016). We consistently detected significantly smaller and fewer tumors in the *bazedoxifene*-treated cohorts compared with the vehicle-treated cohorts (Figs 4B–D and EV1A). Importantly, we observed reduced tumor burden in both, *bazedoxifene*-treated *gp130*$^{Y757F}$ male and female mice, suggesting that the tumor suppressive effect was not mediated through *bazedoxifene*'s SERM activity (Fig 4C and D). We confirmed the latter with our observations that *bazedoxifene* also reduced tumor burden following depletion of estrogen producing cells in a cohort of female ovariectomized mice (Fig EV2). Consistent with this finding, we also were unable to detect expression of ERα, encoded by the *Esr1* gene, in gastric tumors of mice of either gender (Fig 4E). Thus, our observations argue strongly that the anti-tumor activity of *bazedoxifene* on the gastric epithelium occurs independently of its SERM activity.

We then hypothesized that *bazedoxifene* treatment of tumor-bearing mice would induce the same signaling "hallmarks" observed upon treatment of *gp130*$^{Y757F}$ mice with either IL11-Mutein or JAK inhibitors (Stuart *et al*, 2014). Indeed, we detected reduced levels of the activated (i.e., phosphorylated) isoforms of the signaling proteins STAT3 and Akt (Fig 4F). Akin to our findings with IL11-Mutein therapy (Putoczki *et al*, 2013; Stuart *et al*, 2014), tumors from *bazedoxifene*-treated mice had reduced expression of the pro-survival protein Bcl-x$_L$, of the proliferative marker cyclin D$_1$ and the cancer cell-specific mitosis marker survivin (Fig 4F). These observations correlated with a significant decrease in Ki67-stained tumor epithelium and an increased number of TUNEL-positive, apoptotic cells in tumors from *bazedoxifene*-treated mice (Fig EV1B). Furthermore, expression of mRNA transcripts for the *bona fide* STAT3-target genes *Socs3*, *Icam1*, and *Reg3a*, which are both induced by IL11 in gastric tumors (Putoczki *et al*, 2013), were significantly decreased in tumors from *bazedoxifene*-treated animals (Fig 4G). Collectively, our data support a mechanism of action whereby *bazedoxifene* directly impairs intracellular IL11 signaling.

### Bazedoxifene reduces *Apc*-dependent colon cancer growth *in vivo*

We and others have previously shown that IL11 signaling, and to a lesser extent IL6 trans-signaling, enables and promotes the formation and progression of epithelial tumors that arise in the intestinal epithelium in response to aberrant canonical WNT signaling associated with either loss-of-function mutations in the *Apc* tumor suppressor gene or gain-of-function mutations in β-catenin (Becker *et al*, 2004; Ernst *et al*, 2008; Putoczki *et al*, 2013). We surmised from these observations that gp130-mediated STAT3 signaling conferred a rate-limiting, permissive signaling for survival and proliferation of intestinal epithelium harboring *bona fide* mutations in these WNT signaling genes (Phesse *et al*, 2014).

To explore whether *bazedoxifene* could also curb *Apc*-dependent colon cancer development, we exploited *Cdx2*$^{CreERT2}$; *Apc*$^{flox}$ mice, which develop tumors within 2 weeks following *tamoxifen*-dependent bi-allelic inactivation of the *Apc* gene in the epithelium of the colon and cecum (Hinoi *et al*, 2007; Fig 5A). We treated tumor-bearing *Cdx2*$^{CreERT2}$; *Apc*$^{flox}$ mice with *bazedoxifene* for a 2 week duration, with treatment commencing 2 weeks after *tamoxifen*

administration, and observed a significant reduction in overall tumor burden (Fig 5B–F). Akin to our observation with the *gp130*$^{Y757F}$ model, *bazedoxifene* treatment conferred a comparable anti-tumor effect in male and female *Cdx2*$^{CreERT2}$; *Apc*$^{flox}$ mice (Fig 5B and E). As the therapeutic effect of *bazedoxifene* treatment reduced tumor number (Fig 5C) as well as tumor size (Fig 5D), we inferred that *bazedoxifene* treatment suppressed both tumor initiation (affecting tumor numbers) and progression (affecting tumor size). This result is consistent with our previous findings for a role for STAT3 signaling in models of *Apc*-dependent colon tumorigenesis (Phesse *et al*, 2014). Western blot analysis of colon tumors confirmed strongly reduced levels of pSTAT3 in *bazedoxifene*-treated mice, coinciding with reduced Bcl-x$_L$, cyclin D1, and survivin expression (Fig 5G). We corroborated these findings by immunohistochemical analysis of tumor sections, which revealed reduced pSTAT3 staining in tumor epithelium that coincided with reduced proliferation and increased apoptosis as revealed by staining for Ki67 and TUNEL, respectively (Fig EV3A). Finally, we used qPCR analysis to confirm reduced expression of the *bona fide* STAT3-target genes *Socs3*, *Icam1*, *Reg3a,* and *Bmi-1* in tumors of *bazedoxifene*-treated mice (Fig 5H), but not of the β-catenin target genes *Sox9*, *axin2*, and *lgr5* (Fig 5I). To confirm that *bazedoxifene* treatment reduced the growth of colon cancer cells, we also derived organoids from tumors of *Apc*$^{Min}$ mice and observed that *bazedoxifene* treatment reduced IL11-dependent organoid growth (Fig EV3B).

We next ascertained that the striking anti-tumor effect of *bazedoxifene* treatment also extended to the epithelium of the small intestine, which is also susceptible to the therapeutic benefit conferred by inhibition of the IL11/STAT3 signaling axis at the level of the gp130-associated JAK1/2 kinases (Phesse *et al*, 2014). For this, we induced intestinal tumors using the stem cell marker locus *Lgr5* to confer bi-allelic *Apc* deletion following *tamoxifen* treatment of *Lgr5*$^{CreERT2}$; *Apc*$^{flox}$ mice. We observed that administration of *bazedoxifene* for 2 weeks to tumor-bearing *Lgr5*$^{CreERT2}$; *Apc*$^{flox}$ mice also significantly reduced overall tumor burden in the small intestine (Fig 6A).

Consistent with our observations in the *Cdx2*$^{CreERT2}$; *Apc*$^{flox}$ colon cancer model, immunohistochemical analysis of tumors from *bazedoxifene*-treated *Lgr5*$^{CreERT2}$; *Apc*$^{flox}$ mice revealed a reduction in nuclear pSTAT3 and Ki67 staining that coincided with augmented TUNEL staining (Fig EV4). Tumors from *bazedoxifene*-treated *Cdx2*$^{CreERT2}$; *Apc*$^{flox}$ and *Lgr5*$^{CreERT2}$; *Apc*$^{flox}$ mice also showed reduced levels of the STAT3-target gene *bmi-1* (Fig 5H and data not shown), encoding a polycomb repressor component that suppresses expression of cell cycle inhibitors, and therefore restricts intestinal tumor growth in *Lgr5*$^{CreERT2}$; *Apc*$^{flox}$; *bmi-1*$^{+/-}$ mice (Fig 6A; Phesse *et al*, 2014). Collectively, our data suggest that the anti-tumor mechanism of action of *bazedoxifene* occurs via inhibition of STAT3 signaling and results in suppressed cell survival and proliferation of *Apc*-mutagenized intestinal epithelium.

### Aberrant WNT signaling is retained in *Apc*-mutant tumors of bazedoxifene-treated mice

To rule out that *bazedoxifene* directly interferes with canonical WNT signaling in tumors of *Lgr5*$^{CreERT2}$; *Apc*$^{flox}$ mice, we used immunohistochemistry to detect transcriptionally active surrogates of canonical WNT signaling (nuclear β-catenin) and gp130 signaling

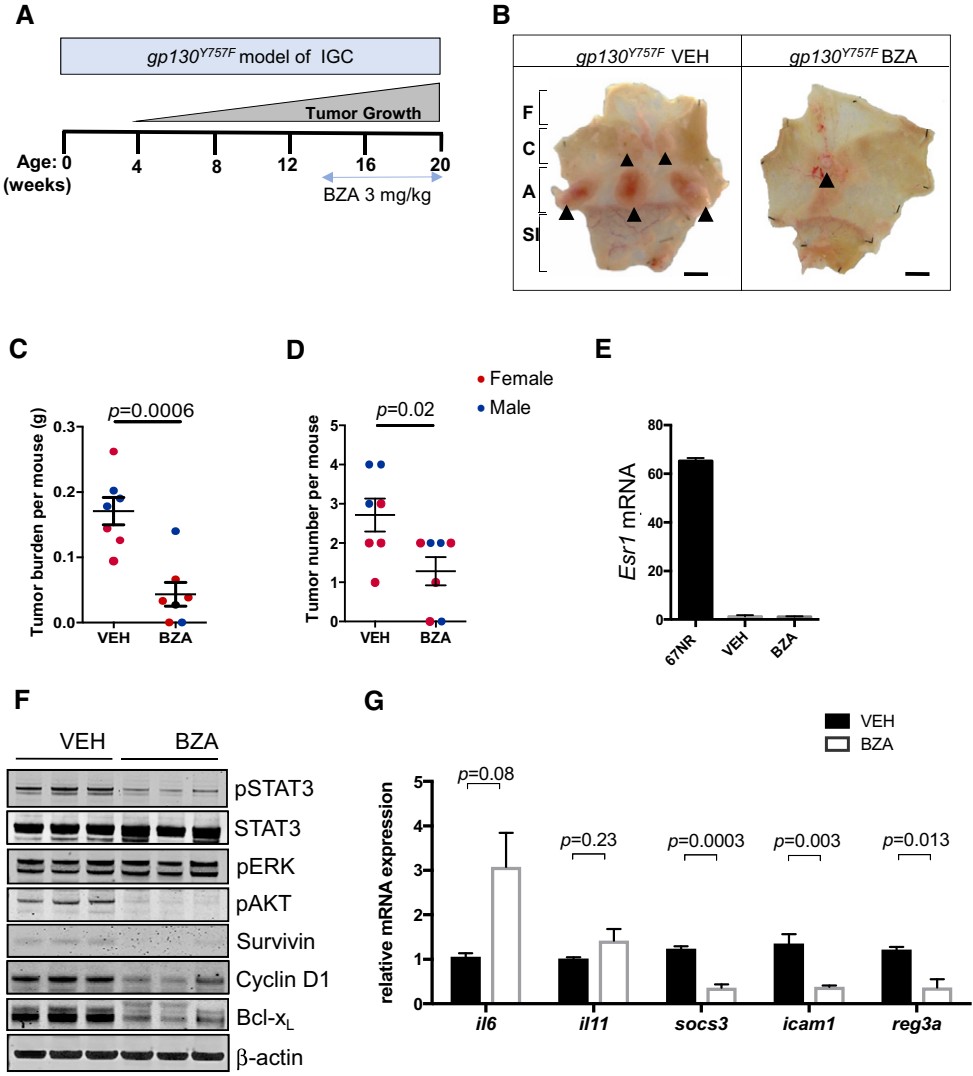

**Figure 4.** *Bazedoxifene* treatment inhibits growth of endogenous arising gastric tumors.

A     Schematic representation of experimental setup. Intestinal-type gastric cancer occurs spontaneously in homozygous $gp130^{Y757F}$ mice from 4 week of age with 100% penetrance, and tumor-bearing mice were treated with *bazedoxifene* (BZA) between weeks 13–20 at 3 mg/kg.

B     Representative whole mounts of pinned-out stomachs and opened along the outer curvature, from mice treated with vehicle (VEH) or BZA as described in (A). Tumors are indicated by arrow heads. F, fore stomach; C, corpus; A, antrum; SI, small intestine. Scale bar = 5 mm.

C, D   Total tumor burden (C) and multiplicity (D) determined in individual 20-week-old mice from vehicle- and BZA-treated cohort as indicated in (A). Each symbol represents an individual mouse of the indicated gender. Data are mean ± SEM, n = 7, exact *P*-values from unpaired Student's *t* test.

E     ERα mRNA (*esr1*) levels in tumors collected from the gastric antrum of mice from the vehicle- and BZA-treated cohorts. ERα expression in the mammary tumor cell line 67NR is provided as a positive control.

F     Western blot analysis of tumor lysates from tumors collected in (B) for the activated (phosphorylated) isoforms of signaling proteins as well as for the STAT3 target genes survivin, Bcl$_{XL}$, and cyclin D1. Each lane corresponds to tumors pooled from an individual mouse. Blots were reprobed for β-actin as a loading control.

G     Changes in mRNA expression of IL11-regulated STAT3 target genes in tumors from vehicle and BZA-treated mice. Expression is normalized to mean value of vehicle-treated mice. Data are mean ± SEM, n = 5, *P*-values obtained from unpaired Student's *t* test.

Source data are available online for this figure.

(pSTAT3) in tumor epithelium from mice in the *bazedoxifene*- and the vehicle-treated cohorts. While we detected comparable signals for β-catenin, pSTAT3 staining was reduced selectively in tumors collected from mice in the *bazedoxifene*-treated cohort (Fig EV4). We functionally confirmed this observation in IL11-stimulated human SW480 colon cancer cells that harbor homozygous impairment mutations in the *APC* gene and therefore show aberrant WNT

signaling. For this, we transfected SW480 cells with either the pTOP-FLASH or pAPRE-luc reporter plasmids to specifically record β-catenin- and STAT3-dependent transcriptional activity, respectively. This analysis revealed that *bazedoxifene* treatment did not suppress aberrant β-catenin-dependent pTOPFLASH activity (Fig 6B), but antagonized IL11-induced pAPRE-luc reporter activity (Fig 6C). Indeed, the effect of *bazedoxifene*-dependent inhibition of pAPRE-

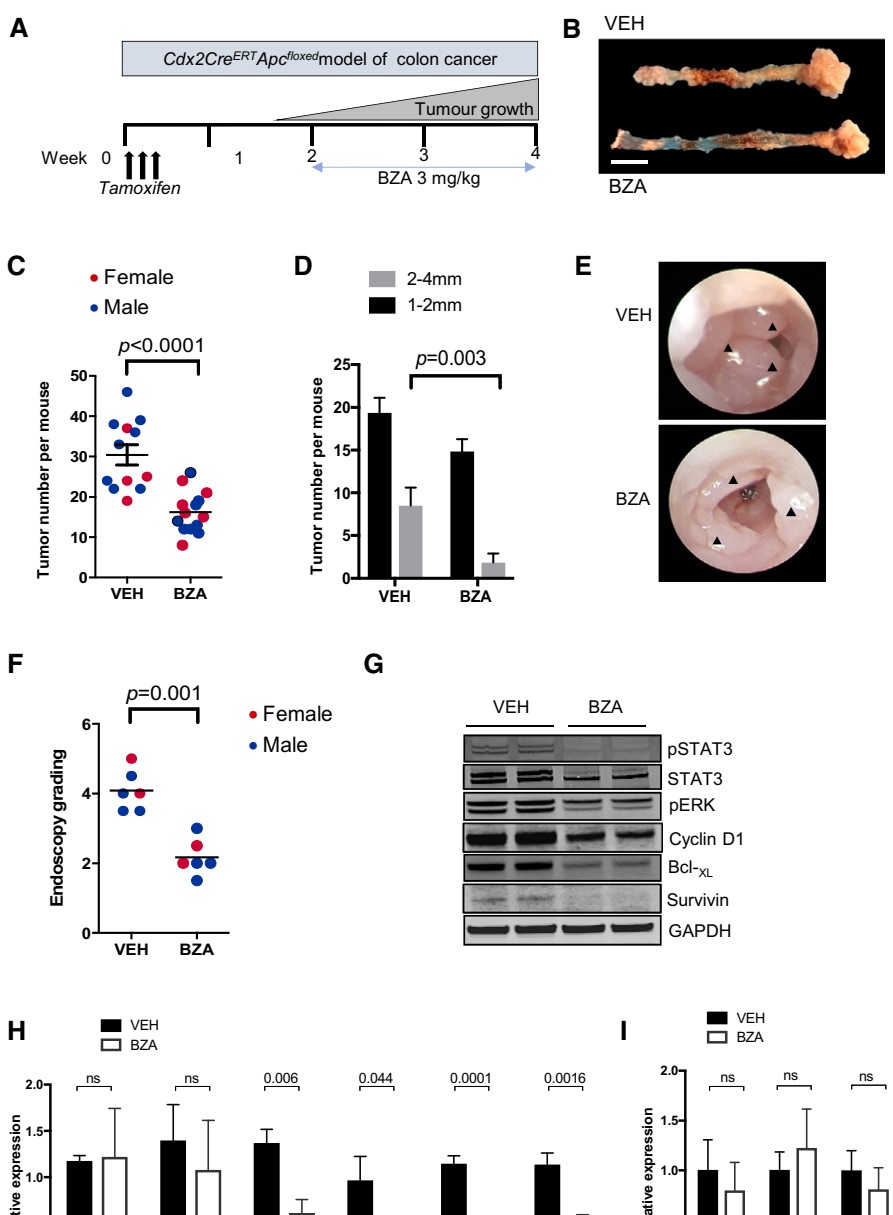

**Figure 5. *Bazedoxifene* treatment inhibits growth of endogenous arising colon tumors in *Cdx2*ᶜʳᵉᴱᴿᵀ²; *Apc*ᶠˡᵒˣ mice.**

A Schematic representation of experimental setup. Colon cancer occurs in adult *Cdx2*ᶜʳᵉᴱᴿᵀ²; *Apc*ᶠˡᵒˣ mice 10–14 days after *tamoxifen* administration (1 mg/day for 3 consecutive days) and associated induction of Cre recombinase activity to confer bi-allelic inactivation of the *Apc* tumor suppressor gene. *Bazedoxifene* (BZA) treatment was initiated 14 days post-*tamoxifen* administration and lasted for two consecutive weeks.

B Representative examples of colons (left) and cecum (right) of vehicle- (VEH) and BZA-treated mice as described in (A). Scale bar = 10 mm.

C, D Quantification of total tumor number in the colon of individual mice of the indicated gender (C), and when classified according to the size of individual tumors (D) at the end of 2 weeks of BZA treatment (mean ± SEM, $n = 12$ untreated controls, $n = 13$ BZA-treated group, unpaired Student's *t* test). Each symbol represents an individual mice of the indicated gender.

E, F Representative endoscopy images of tumor-bearing mice 2 weeks after vehicle or BZA treatment. Arrowheads indicate tumors (E). Endoscopy score of tumor burden as described in the Materials and Methods section (F) (Unpaired Students *T* test).

G Western blot analysis of tumor lysates from (C) for the activated (phosphorylated) isoforms of signaling proteins, as well as for the STAT3 targets survivin, Bcl$_{XL}$ and cyclin D1. Each lane corresponds to tumors pooled from an individual mouse.

H, I Changes in mRNA expression of target genes regulated by STAT3 (H) or the canonical WNT/β-catenin pathway (I) in vehicle- and BZA-treated mice. Expression is normalized to mean value of vehicle-treated mice. Data are mean ± SEM, $n = 5$ per group, $P$-values from unpaired Student's *t* test.

Source data are available online for this figure.

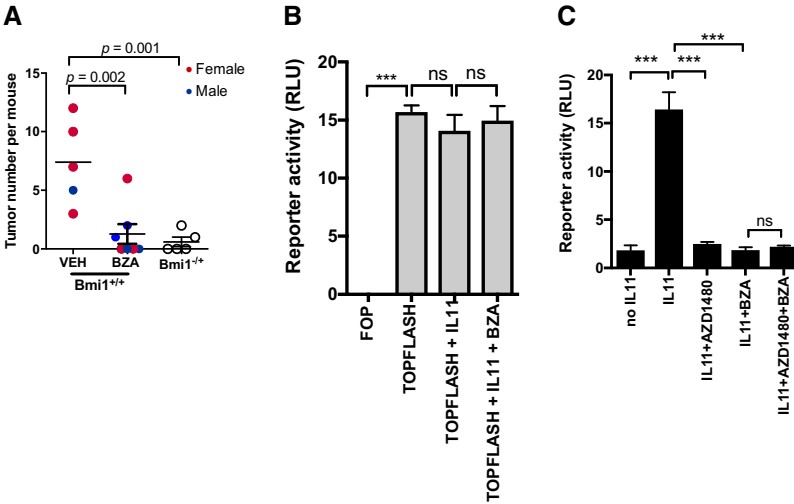

**Figure 6. *Bazedoxifene* treatment suppresses tumor growth through a STAT3-dependent mechanism.**

A    Quantification of tumor numbers in the small intestinal of adult $Lgr5^{CreERT2}$; $Apc^{flox}$ mice occurs 10–14 days after tamoxifen administration (1 mg/day for 3 consecutive days) and the associated induction of Cre recombinase activity to confer bi-allelic $Apc$ inactivation. *Bazedoxifene* (BZA) treatment (3 mg/kg, 3 times per week) was initiated 14 days post-*tamoxifen* and lasted for 2 consecutive weeks. Mono-allelic $Bmi1$ ablation reduces tumor burden in $Lgr5^{CreERT2}$; $Apc^{flox}$;$Bmi1^{+/-}$ mice when compared to their fully $Bmi1$-proficient littermates. Each symbol represents an individual mice of the indicated gender. Data are mean ± SEM, $n = 5$ untreated controls, $n = 7$ BZA-treated group of $Lgr5^{CreERT2}$; $Apc^{flox}$ mice and $n = 5$ $Lgr5^{CreERT2}$; $Apc^{flox}$;$Bmi1^{+/-}$, P-values obtained from unpaired Student's t test).

B    Assessment of canonical WNT/β-catenin signaling in SW480 human colon cancer cells which harbor bi-allelic $APC$ impairment mutations and were transfected with either the firefly luciferase pTOPFlash positive or the negative control pFOPFlash control reporter construct alongside a non-responsive Renilla luciferase plasmid. Cells were then treated for 24 h with IL11 (10 ng/ml) and BZA (1 μM) as indicated. Results are expressed as relative luciferase units (RLU), that is, firefly luciferase activity normalized against Renilla luciferase activity in each individual culture. Data are mean ± SEM, $n = 3$, ***$P = 0.0001$, ns = not significant, ANOVA with Dunnett's comparisons test.

C    Assessment of STAT3 signaling in SW480 human colon cancer cells transfected with the pAPRE reporter and the non-responsive Renilla-luc construct. Cells were then treated for 24 h with IL11 (10 ng/ml), BZA (1 μM), or the Jak1/2 inhibitor AZD1480 (1 μM) as indicated. Results are expressed as RLU, that is, firefly luciferase activity normalized against Renilla luciferase activity in each individual culture. Data are mean ± SEM, $n = 3$, ***$P = 0.0001$, ns = not significant, ANOVA with Dunnett's comparisons test.

luc activity was similar to that observed in SW480 cells treated with the JAK1/2 inhibitor AZD1480 (Fig 6C).

Collectively, our observations derived in several independent, but complementing mouse and human-derived model systems provide orthogonal evidence for the *in vivo* effects of *bazedoxifene* in suppressing the growth of $Apc$-driven cancers, most likely through a process involving the inhibition of IL11 signaling in epithelial tumor cells.

### *Bazedoxifene* treatment does not alter tumor immune responses

While our data thus far argue strongly in favor of *bazedoxifene* conferring anti-tumor activity directly on epithelial cells via a STAT3-dependent, but SERM-independent mechanism, our next aim was to assess whether the anti-tumor activity of *bazedoxifene* was conferred in part by systemic, immune cell-dependent mechanisms. For this, we extensively profiled tumor infiltrating immune cells in $gp130^{Y757F}$ (data not shown) and in $Cdx2^{CreERT2}$; $Apc^{flox}$ mice treated with vehicle or *bazedoxifene* (Appendix Figs S3 and S4). Our flow cytometry-based immune cell profiling included the quantitation of $CD4^+/CD8^+$ effector T cell ratios, exhaustion of $CD8^+$ T cells, myeloid-derived suppressor cells (MDSC), and tumor-associated macrophages. This analysis demonstrated neither changes in $CD4^+$ and $CD8^+$ T cells numbers, nor in the proportion of PD-1$^{high}$ exhausted $CD8^+$ T cells between vehicle or *bazedoxifene*-treated cohorts (Appendix Fig S4). Similarly, we did not detect differences

in either $CD45^+$, $CD11b^+$; $Ly6C^{hi,}$ $Ly6G^{lo}$ monocytic, or $CD45^+$, $CD11b^+$, $Ly6C^{lo}$, $Ly6G^{hi}$ granulocytic MDSCs, nor in $F4/80^+$ tumor-associated macrophages between the two cohorts (Appendix Fig S5), despite reports that some SERMs may affect neutrophils and decrease lymphopoiesis (Bernardi *et al*, 2015; Nordqvist *et al*, 2017). Collectively, these data suggest that the anti-tumor effect of *bazedoxifene* is primarily mediated by tumor cell-intrinsic mechanism rather than by altering the host's anti-tumor immune response.

## Discussion

Here, we provide the first proof-of-concept study demonstrating SERM-activity independent, anti-tumor effects of *bazedoxifene* in a comprehensive series of preclinical mouse models for gastrointestinal cancers. We identify interference with IL11 binding to the gp130 receptor signaling subunit as the mechanism by which *bazedoxifene* suppresses tumor growth in male and females hosts and on cells lacking ERα expression. Akin to our observations that inhibition of gp130 signaling at the level of ligand binding, engagement of the gp130-receptor-associated JAK kinases or ultimately activation of the STAT3 signal transducer suppresses tumor formation in $gp130^{Y757F}$, $Lgr5^{CreERT2}$; $Apc^{flox}$ and $Cdx2^{CreERT2}$; $Apc^{flox}$ mice; *bazedoxifene* treatment replicated these effects. Notably, *bazedoxifene* also suppressed tumor growth in mutant $Apc$-driven intestinal

tumors and in human colon cancer cell lines without reducing the excessive signaling output of the canonical WNT pathway that constitutes the underlying oncogenically activated driver pathway. Accordingly, *bazedoxifene* treatment also reduced the growth of patient-derived colon cancer organoids that harbor the *APC* driver, and possibly compounding additional mutations. However, given that conventional IL11 signaling and IL6 trans-signaling both facilitate the growth of intestinal tumors (Putoczki *et al*, 2013; Schmidt *et al*, 2018), our observations do not address the relative contribution by which the *bazedoxifene* effect is mediated through inhibition of each of these mechanisms. Given the collective contribution of these gp130 cytokines in driving tumorigenesis, *bazedoxifene* may represent a potent mode for the simultaneous inhibition of conventional IL11 and IL6 trans-signaling in tumors.

Elevated levels of IL6, IL11, their cognate receptor α-subunits, and associated signaling activity, as assessed by the abundance of the activated pSTAT3 isoform, are commonly observed in colon, gastric, and other solid malignancies and often correlate with increased metastasis and poorer patient outcomes (Kang *et al*, 2003; Putoczki *et al*, 2013; Johnstone *et al*, 2015). In our study, tumor epithelial cells isolated from stage II primary colon cancers, mRNA transcript expression of IL6 and IL11, their cognate receptors, and gp130 were detected although it was the expression levels of gp130 which determined cellular sensitivity to IL6 and IL11 stimulation. This suggests that expression levels of gp130 may be relevant as a biomarker in patient selection for targeted anti-IL6 or anti-IL11 treatments. An anti-tumorigenic effects of *bazedoxifene*, associated with reduced IL6-dependent pSTAT3 levels, have been demonstrated in xenograft models of rhabdomyosarcoma (Xiao *et al*, 2017), pancreatic (Wu *et al*, 2016), liver (Wang *et al*, 2017), and colon cancer (Li *et al*, 2018). Our data comprehensively extend these observations and demonstrate significant anti-tumorigenic effects of *bazedoxifene* in multiple IL11-dependent and immune-competent models of spontaneous gastric, intestinal, and colon cancer.

*Bazedoxifene* is an indole-based ER ligand with unique structural characteristics not shared with anti-estrogens *tamoxifen* and *raloxifene*. Although *bazedoxifene* binds to ERα with greater affinity than to ERβ, *bazedoxifene* is a less selective ERα binder than *raloxifene* from which its structure was derived (Wardell *et al*, 2013). Upon the identification of *raloxifene* and *bazedoxifene* as potential gp130-interacting compounds, Li *et al* (2014) proposed a mechanism whereby IL6 inhibition is likely to result from the disruption of hexameric IL6:IL6Rα:gp130 complex formation. Mutagenesis studies demonstrated that tryptophan-157 in IL6 was the most critical residue at the Site III interface (Barton *et al*, 1999; Boulanger *et al*, 2003; Veverka *et al*, 2012). Meanwhile, our crystallographic analysis of IL11, IL6, and the IL6 hexameric signaling complex suggests that the two cytokines interact in a similar manner through Site III with gp130 and show functional conservation between tryptophan-157 in IL6 and tryptophan-168 in IL11 (Putoczki *et al*, 2014). These insights also predict our findings that *bazedoxifene* should not affect LIFR signaling as the corresponding signaling-competent trimeric LIF:LIFR:gp130 receptor complex does not rely on Site III-dependent interactions (Huyton *et al*, 2007). Likewise, the models also foreshadows our experimental observations here that *bazedoxifene* is unable to interfere with signaling from the chimeric L-gp130 molecule, in which the entire extracellular domain is replaced with a leucine zipper.

Besides the beneficial effect of *bazedoxifene* on bone mineral density and risk reduction for new vertebral fractures in postmenopausal women (Silverman *et al*, 2012; Palacios *et al*, 2015; Peng *et al*, 2017), *bazedoxifene* has also been reported to reduce the serum concentrations of total and low-density lipoprotein cholesterol (Stevenson *et al*, 2015). The well-documented antitumor effects of *bazedoxifene* observed in models for ovarian (Romero *et al*, 2012) and *tamoxifen*-resistant breast cancer (Wardell *et al*, 2013) have been mechanistically attributed to the capacity of *bazedoxifene* to compete with 17β-estradiol for binding to ERα in these steroid-sensitive tissues. The evidence collectively presented here strongly suggests that the therapeutic effect of *bazedoxifene* observed on gastrointestinal cancer occurs independent of ERα signaling, as ERα expression in the gastrointestinal epithelium was below detection level. Functionally even more important, *bazedoxifene* conferred similar therapeutic benefits to male and female mice in all of our *in vivo* models and also reduced gastric tumor burden in ovariectomized female $gp130^{Y757F}$ mice. Likewise, the anti-tumor activity of *bazedoxifene* in our gastrointestinal cancer models was also not mediated by systemic effects through tumor-associated immune cell subsets even though some SERMs may affect granulopoiesis and lymphopoiesis (Nordqvist *et al*, 2017).

Treatment of postmenopausal osteoporosis in women with *bazedoxifene* is approved by the FDA at doses in the range of 20–40 mg daily. Accounting for the pharmacokinetic difference between humans and mice, the FDA-recommended daily intake of 20 mg *bazedoxifene* for a 60 kg woman corresponds to a dose of approximately 4.1 mg/kg/day in mice (Nair & Jacob, 2016). The latter dose is consistent with the 3 mg/kg dose administered here and in some of the previous studies documenting the anti-estrogen activities of *bazedoxifene* in mice (Sakr *et al*, 2014; Flannery *et al*, 2016). Thus, the safety profile of *bazedoxifene* may confer benefits to a large (aging) population. Besides protecting against bone loss and beneficially affecting cholesterol metabolism (Silverman *et al*, 2012; Stevenson *et al*, 2015), here we provide evidence in controlling the growth of adenomatous lesions, where mutations in *APC* form the initial oncogenic insult in 80% of all metastatic colon cancers. By exploiting the capacity of *bazedoxifene* to inhibit those inflammatory cytokines that most prominently promote the growth of gastrointestinal tumors, our data provide a compelling argument for the repurposing of *bazedoxifene* for a potentially novel treatment modality for gastrointestinal cancers.

# Materials and Methods

### Cell culture and reporter activity assays

Cell lines were maintained at 37°C in 10% $CO_2$ with appropriate culture media supplemented with 10% (v/v) of fetal calf serum (FCS) and 1% (v/v) of GlutaMAX (Gibco). LIM2405 and MDA-MB231 cell lines in Dulbecco's modified Eagle's medium (DMEM)/F-12; HEK293, MKN1, $BAF/03^{IL6R\alpha}$, $BAF/03^{IL11R\alpha}$ and $BAF/03^{LIFR}$ cell lines in RPMI1640. Transfections were performed with Lipofectamine 2000 according to manufacturer's instructions (Invitrogen). To transfect HEK293 cells, 150,000 cells were plated in 6-well plates and transfected with 100 ng of pAPRE-luciferase, 100 ng of pcDNA-

hIL11RA, and 100 ng of pRL-TK-Renilla Luciferase expression constructs, with a 3:1 Lipofectamine: DNA ratio. Cells were serum starved 1 day post-transfection and treated with IL11 (10 ng/ml) and *bazedoxifene* (ranging from 0.003 to 3 μM, Selleck Pharmaceuticals) 2 days post-transfection. Luciferase assays were performed using the dual luciferase reporter kit (Promega) using manufacturer's instructions. Firefly luciferase readings were normalized to Renilla activity. TOPFLASH and FOPFLASH Reporter assays were performed in SW480 cells using previously published methods (Phesse *et al*, 2014).

### BAF/03 cytokine-dependent cell proliferation assays

Transfections in BAF/03 cells were performed with the Amaxa Nucleofector (Lonza) with 1 μg plasmid DNA (pcDNA-IL11RA and pcDNA-L-gp130). Parental BAF/03 cells were maintained in 10 ng/ml of IL3, and BAF/03$^{IL6Rα}$, BAF/03$^{IL11Rα}$, and BAF/03$^{LIFR}$ were maintained in 10 ng/ml of IL3 and 10 ng/ml of IL6, IL11, or LIFR, respectively. Cell proliferation was measured at Day 0 upon treatment initiation and at 24 h post-treatment. Cell proliferation rates were measured on Day 0 upon the initiation of treatment and after 24 h of treatment. Changes in cell proliferation were determined relative to the mean values obtained from cytokine and vehicle co-treated wells. Data are expressed as fold change. Cell proliferation was determined with the MTT (3-(4,5-dimethylthiazol-2-yl)-2,5-diphenyltetrazolium bromide) assay according to manufacturer's instructions with the measurement of absorbance at 490 nm (Thermo Fisher; M6494).

### Structural modeling

*Bazedoxifene* was manually docked to gp130 Site III (PDB ID:1P9M) using SYBYL-X 2.1.1 (Certara, L.P. Princeton, NJ, USA) to verify the binding modes previously published. To remove any steric issues that may have arisen during the docking process, the bazedoxifene ligand and gp130 residues within an 8 Å radius of the docked ligand were geometrically optimized within SYBYL-X 2.1.1 using the MMFF94s force field and partial atomic charges, the conjugate gradient convergence method and 10,000 iterations. To illustrate the structural differences between the two compounds, *tamoxifen* and *bazedoxifene* were aligned via their common phenoxypropylamine core. The PyMOL Molecular Graphics System, version 1.8.2.2 (Schrödinger LLC, Cambridge, MA; http://www.pymol.org), was used for all structural visualization, interaction analysis, and figure creation.

### Patient-derived colon cancer primary cell isolation and culture

Patient-derived colon cancer cells were derived from resection specimens collected from Stage II CRC patients using a previously described protocol (Grillet *et al*, 2017), under agreement from the Centre Hospitalier Universitaire de Nimes Human Ethics Committee (#2011-A01141-40). Informed consent was obtained from all subjects and experiments performed in accordance with the WMA Declaration of Helsinki and the Department of Health and Human Services Belmont Report. Briefly, tumor samples were incubated for 5 min in HBSS + 0.4% bleach, washed five times in HBSS, minced into 2-mm$^3$ fragments, and dissociated using liberase H

(0.26 U/ml, Roche) diluted in Accumax (Sigma-Aldrich) for 2 h at 37°C. Thereafter, isolated cells were filtered through a 40-μm mesh and single-cell suspensions were plated in DMEM, supplemented with FBS, glutamine, 100 μg/ml gentamicin, and 40 μg/ml ciprofloxacin.

### Human cancer organoid culture and treatment

Human cancer cultures were established and maintained as described previously with modifications (Jung *et al*, 2011). For assay setup, the passaged organoids were harvested and broken down into small fragments mechanically using a 26-G needle. The fragments were mixed with Matrigel (BD Biosciences #356231) and plated mechanically using an Epmotion Automated workstation 5073c (Eppendorf) into 384-well plates (Corning #3985). Approximately 200 fragments per well were embedded in 8 μl Matrigel and seeded in 384-well plates. The Matrigel was allowed to polymerize at 37°C for 60 min after which 80 μl of growing media was dispensed into each well using the Epmotion 5073c. On Day 6 of the culture, the Epmotion Automated workstation 5073c (Eppendorf) was used to replenish media with treatments. The experiment used in the study was terminated on Day 13 (Day 7 after treatment). The cultures were monitored by microscopy throughout the experiment where bright-field image stacks were acquired of each used well on most days.

### Human cancer organoid image acquisition, processing, and scoring

The detailed image acquisition and analysis procedures for acquiring images in 96-well plates have been described previously (Hirokawa *et al*, 2014; Tan *et al*, 2015). In this experiment, 384-well plates (Corning #3985) were used. The microscopy was conducted using a Nikon Eclipse Ti-U microscope with a 4× objective lens and motorized stage (Prior Scientific, H117). Bright-field image stacks of each well were captured on a Nikon DS-Ri2 camera (Nikon Inc, MQA17000) using the NIS-Elements software (Nikon, Basic Research version 4.40). The depth images of each FOV (well) were then stacked using the EDF algorithm in Fiji. The processed images were analyzed semi-automatically using a customized Fiji-script where objects identified were bordered a region of interest (ROI) and these ROIs were tabulated and analyzed. The resulting ROIs were manually curated, and the objects sizes and mean intensities acquired.

### Animal experiments

All animal experiments were conducted in accordance with ethics protocols approved by Austin Health and La Trobe University Animal Ethics Committees. Transgenic C57BL/6 *gp130*$^{Y757F}$, *Lgr5*$^{CreERT2}$; *Apc*$^{flox}$ and *Cdx2*$^{CreERT2}$; *Apc*$^{flox}$ mice were bred and maintained under specific pathogen-free housing conditions. Mouse housing conditions included a constant temperature of 24°C with a 12-h light/dark cycle, 5 mice per cage with food and water *ad libitum* as described previously (Tebbutt *et al*, 2002; Hinoi *et al*, 2007; Phesse *et al*, 2014). Experiments were performed with mice of both genders, 9–10 weeks of age in all experiments except for those with *gp130*$^{Y757F}$ mice, which were used at 13 weeks of age. To induce

Cre-mediated exon deletion, a single-daily intra-peritoneal (i.p.) injection of tamoxifen was administered over three consecutive days in the *Lgr5*^CreERT2; *Apc*^flox and *Cdx2*^CreERT2; *Apc*^flox mice. *Bazedoxifene* (3 mg/kg, Selleckchem) dissolved in 10% v/v dimethyl sulfoxide (DMSO) and 90% v/v sunflower oil daily was i.p. administered to animals for 5 days of the week, for a period of 7 consecutive weeks in *gp130*^Y757F, and 2 weeks in *Lgr5*^CreERT2; *Apc*^flox and *Cdx2*^CreERT2; *Apc*^flox mice. Vehicle-treated control cohorts were administered 100 μl of 10% v/v DMSO and 90% v/v sunflower oil daily for the same duration. All experiments were performed in age-matched mice. Tumor onset and progression in the distal colon, in *Cdx2*^CreERT2; *Apc*^flox mice, were monitored by endoscopy as previously described (Putoczki *et al*, 2013). Mice were euthanized by $CO_2$, and tissue was collected for formalin fixation for immunohistochemistry and snap-frozen for Western blot and qPCR analyses.

## Ovariectomy

Female *gp130*^Y757F mice were anesthetized using 4% isoflurane, and a small (1 cm) transversal (vertical) incision was made in the skin in the center of the lower back and dissected from the underlying fascia to reach the abdominal wall. A small (5 mm) incision was made in the abdominal wall and the adipose tissue surrounding the ovary to expose the ovary and connected uterine horn. The vessel between the ovary and uterine horn was ligated, and the ovary was dissected out. The uterine horn was then replaced into the abdominal cavity, and the procedure was then repeated for the other side. The skin wound was then closed and secured with a sterile surgical clip. The mice were placed in a heated cage and closely monitored for 2 h.

## Gene expression analysis

Quantitative reverse-transcriptase PCR was performed on RNA isolated from cell lines, tumours, and whole stomach, small intestine, and colon lysates. RNA extraction was performed using the ReliaPrep RNA Cell Mini-Prep System (Promega) according to the manufacturer's instructions. First-strand cDNA synthesis using 1.0 μg total RNA was performed using SuperScript® III First Strand Kit (Life Technologies) primed by random hexamer primers. mRNA was quantified using primer sequences (Appendix Table S1) using the SensiMix SYBR (Bioline) on the 7500 Fast Real Time PCR System (Applied Biosystems). Relative expression was calculated using the comparative CT method ($2^{-ddCt}$) method after normalization to *Hprt* as the housekeeping control gene (Coulson *et al*, 2017). Comparative real-time PCR results were performed in triplicates for all samples.

## Western blot analysis

Protein lysates containing a total of 30 μg of protein, 25% NuPAGE LDS sample buffer, and 10% NuPAGE reducing agent were heat-denatured and separated on precast 4–12% NuPAGE Bis-Tris mini gels (Thermo Fisher Scientific). The SeeBlue Plus2 Protein Standard ladder was used to determine protein band sizes (Thermo Fisher Scientific). Proteins were transferred to PVDF or nitrocellulose membranes using iBlot™ Gel Transfer Stacks for 7 min with 23V using the iBlot Gel Transfer Device (Thermo Fisher Scientific). The

PVDF or nitrocellulose membranes with transferred proteins were then blocked with Odyssey blocking buffer (TBS; Li-COR) for 1 h at room temperature. Membranes were incubated overnight at 4°C with primary antibodies diluted in solution of Odyssey blocking buffer (TBS) and TBST (1:1). After washes in TBST, secondary antibody incubations were performed at room temperature for 1 h (Appendix Table S2). Following washing in TBST, the membranes were visualized using the Licor Odyssey Imaging system.

## Immunohistochemistry

Immunohistochemistry was performed using antibodies directed against pSTAT3, Ki67, and β-catenin as previously published (Putoczki *et al*, 2013; Phesse *et al*, 2014; Coulson *et al*, 2017). 4-μm-thick sections from paraffin blocks were re-hydrated, antigen retrieval performed with citrate buffer (0.1%) by microwaving for 15 min. Sections were hydrogen peroxidase-quenched for 20 min ($H_2O_2$, 3%) and blocked for 30 min in TBST supplemented with 10% goat serum and incubated overnight at 4°C with primary antibody (Appendix Table S2). Sections were incubated with biotinylated secondary antibody (1:1,000) for 1 h at room temperature and subsequently incubated with Vectastain Elite ABC HRP Kit (Vector Laboratories). Chromogenic staining was performed using the DAB peroxidase kit, and the sections were counterstained using Mayer's hematoxylin (H-3401, Vector Laboratories). Apoptosis was quantified by the *in situ* apoptosis detection kit, ApopTAG (Millipore) according to the manufacturer's instructions.

Stained sections were imaged with an Aperio Slide Scanner (Leica Biosystems). Whole-tumor sections were selected and immunostaining for β-cadherin, Ki67, pSTAT3, and cleaved caspase 3 were quantified using Aperio ImageScope Positive Pixel Count Algorithm, written specifically for detection of nuclear, cytoplasmic, or membrane immunostains. The algorithm measurements include signal intensity (graded score of 1–3 with increasing signal intensity) and calculation of the fraction of pixels with a positive-color measurement. Immunostaining was measured as a percentage of the total tumor area including epithelial and stromal cells. Data are presented as mean ± SEM, *n* = 5 animals per treatment group. For pSTAT3 immunostaining, data are presented as percentage positive cells stained for signal intensity > 3. Data are presented as percentage positive pixels of the total number of pixels analyzed.

## Flow cytometry

Colons were harvested from animals, and cells were dissociated in digestion buffer (2.5% FCS, 1 mg/ml collagenase III, 0.4 U/ml dispase, and 2 U/ml DNAse in HBSS) for 30 min at 37°C. The cells were incubated in FACS buffer (2.5% of FCS in PBS) with the relevant antibodies for 30 min on ice. The following fluorochrome-conjugated antibodies were used for flow cytometry analysis: Epcam (clone G8.8) #17579180, CD4 (clone GK1.5), F4/80 (clone BM8), Ly6c (clone HK1.4) all from e-Biosciences; CD8 (clone 53-6.7); B220 (clone RA3-6B2), CD11b (clone M1/70), Ly6g (clone 1A8) all from BD Biosciences; PD1 from Invitrogen #46998582; CD45.2 (clone S45015-2) from the WEHI monoclonal antibody laboratory, TCRβ (Biolegend #109208; Appendix Table S3). Dead cells were excluded by using SytoxBlue (Invitrogen). The data were analyzed by using FlowJo software version 10.3 (FlowJo, LLC 2006-2017). Gating

## The paper explained

### Problem

Colon and stomach cancers collectively are among the most common malignancies and arise in the epithelial lining of the gastrointestinal tract. Specifically, gp130 signaling becomes rate limiting for the growth of gastrointestinal tumors arising from oncogenic driver mutations in *Apc*. In the present study, we investigate whether pharmacological targeting of gp130 with the selective estrogen receptor modulator *bazedoxifene*, clinically approved for the treatment of osteoporosis, is an effective treatment in preclinical models of gastric and colon cancers.

### Results

Our *in silico* modeling suggests that the inhibitory activity of *bazedoxifene* is due to its ability to mimic interactions between the Site III interface of gp130 receptor and tryptophan-157 in the IL6, or tryptophan-168 in the IL11 cytokines. *Bazedoxifene* suppresses gastric tumor growth in *gp130*$^{Y757F}$ mice irrespective of their gender, in which tumors arise through excessive IL11-dependent STAT3 signaling. Strikingly, in sporadic colon cancer models based on aberrant activation of the β-catenin/canonical WNT pathway, *bazedoxifene* treatment also reduces tumor burden in mice following conditional ablation of the *Apc* suppressor gene using the *Lgr5*$^{CreERT2}$- or *Cdx2*$^{CreERT2}$-driver alleles. Consistent with our observation that nuclear accumulation of β-catenin remains unaffected in colonic tumors of *bazedoxifene*-treated mice, *bazedoxifene* treatment of human SW480 colon cancer cells harboring mutant *APC* did not affect canonical WNT signaling, but suppressed IL11-dependent STAT3 signaling.

### Impact

Our data suggest that gp130-dependent Stat3 activation is an important and pharmacologically targetable therapeutic opportunity in gastrointestinal cancers.

strategy for the analysis of immune cell populations is shown in Appendix Fig S3.

### Statistical analysis

Data were analyzed using GraphPad PRISM software for statistical analysis with Students' *t* test. Normal distribution was calculated by using the Kolmogorov–Smirnov test. For multiple analysis, statistical significance was calculated using ANOVA, followed by Tukey or Dunnett's *post hoc* test for multiple comparisons where appropriate. A *P*-value < 0.05 was considered significant. Animals were randomized into treatment arms to remove subjective bias; tumor burden was assessed by an investigator with no prior knowledge of treatment details.

**Expanded View** for this article is available online.

### Acknowledgements

We acknowledge the support of the Victorian State Government Operational Infrastructure Support and the National Medical Health and Research Council (NHMRC) of Australia. The authors acknowledge funding support from NHMRC, Cancer Australia, Ludwig Cancer Research, John T Reid Trusts, Cure Brain Cancer Foundation, Cancer Council Victoria, and the Victoria Cancer Agency. The authors acknowledge NHMRC Fellowship support to A.L.C (1062247), M.W.P (1117183), and M.E (1079257), and Australian Research Council Future Fellowship support to M.D.W.G (FT140100544). Authors also acknowledge grant funding from NHMRC to F.H (1069024) and M.E (1092788). We are grateful to Prof Nicola (WEHI) for the kind donation of BAF/03$^{IL6Rα}$ and BAF/03$^{LIFR}$ cells for use in experiments.

### Author contributions

ALC and ME designed research; PT, JH, ARP, CWT, KT, NJH, DB, SA-S, and ALC performed research; TLN, ACP, MB, AMS, MDWG, DB, FH, MWP, TLP, ME, and ALC analyzed data; ALC, TLN, and ME wrote the paper. All authors approved the manuscript.

### Conflict of interest

The authors declare that they have no conflict of interest.

### For more information

**Gp130/IL6ST gene and protein links**

(i)   https://www.ncbi.nlm.nih.gov/gene/3572

(ii)  https://www.omim.org/entry/600694

**IL11 and IL6 gene and protein links**

(i)    https://www.ncbi.nlm.nih.gov/gene/3589

(ii)   https://www.omim.org/entry/147681

(iii)  https://www.ncbi.nlm.nih.gov/gene/3569

(iv)  https://www.omim.org/entry/147620

**Author Website links**

(i)    https://www.onjcancercentre.org/research/research/our-research-programs/cancer-therapeutics-development-group

(ii)   https://www.onjcancercentre.org/profile/ashwini-chand

(iii)  https://www.onjcancercentre.org/research/research/our-research-programs/cancer-inflammation-laboratory

(iv)  https://www.onjcancercentre.org/profile/prof.-matthias-ernst

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
