## [Review Process File · EMBO Molecular Medicine]

Repurposing the selective estrogen receptor modulator bazedoxifene to suppress gastrointestinal cancer growth

Pathum Thilakasiri, Jennifer Huynh, Ashleigh Poh, Chin Wee Tan, Tracy Nero, Kelly Tran, Adam Parslow, Shoukat Afshar-Sterle, David Baloyan, Natalie Hannan, Michael Buchert, Andrew Scott, Michael Griffin, Frederic Hollande, Michael Parker, Tracy Putoczki, Matthias Ernst, Ashwini Chand

Review timeline:

Submission date:	15 July 2018
Editorial Decision:	29 August 2018
Authors' Appeal:	13 September 2018
Resubmission:	28 November 2018
Editorial Decision:	21 December 2018
Revision received:	19 January 2019
Accepted:	21 January 2019

Editor: Céline Carret

Transaction Report:

1st Editorial Decision

29 August 2018

Thank you for the submission of your research manuscript to our editorial office.

We have now received the enclosed reports on it. As you will see, referees 1 and 2 appreciate the repurposing idea of BZA for gastro-intestinal cancers and find the data to be of putative interest. Unfortunately, both referees raise serious concerns regarding the novelty of the findings and conclusiveness of the data and pinpoint several technical issues that preclude a solid interpretation of the experimental evidence provided. In brief, since the effects of BZA have already been shown in animal models of colorectal cancer, we unfortunately feel that the presented work does not progress the translational knowledge enough to justify publication in EMBO Molecular Medicine.

Given the nature of these criticisms and the fact that EMBO Molecular Medicine can only invite revision of papers that receive enthusiastic support from a majority of referees, I am afraid that we do not feel it would be productive to call for a revised version of your manuscript at this stage and therefore we cannot offer to publish it.

We hope that the referees' comments will be helpful to you as you prepare your manuscript for submission elsewhere. We realize that our decision to not pursue publication here will be very disappointing, and we recognize that your findings will be of value to the field.

I would therefore like to propose a transfer of your manuscript and referee reports to the new open-access journal Life Science Alliance. Life Science Alliance is launched as a partnership between EMBO Press, Rockefeller Press, and Cold Spring Harbor Laboratory Press, and publishes work that is of high value to the respective communities across all areas in the life sciences. I already have alerted Dr. Andrea Leibfried, executive editor of Life Science Alliance, to your work, and Andrea would like to consult with her editorial team and get back to you swiftly upon transfer, to confirm their interest and to outline the necessary minor revisions/amendments needed for publication in Life Science Alliance. You can also directly discuss your work with Andrea by contacting her at a.leibfried@life-science-alliance.org

I am very sorry to disappoint you on this occasion and I hope you will view the possibility of a transfer favorably. If this is the case, please use the link below to transfer the manuscript directly.

***** Reviewer's comments *****

Referee #1 (Remarks for Author):

In this paper entitled Repurposing the selective estrogen receptor modulator bazedoxifene to suppress gastrointestinal cancer growth, the authors show the anti-tumor effect of the bazedoxifene (BZA) in gastric/colorectal cancer, based on IL6/IL11 signaling interference, using in vivo experiments in various mouse models.

The in vitro/in vivo treatments with BZA clearly have an effect on tumor cell growth, tumor formation/growth, similar to the effects of interference of IL11 by antagonists or genetic restriction as they have showed before. However, it remains unclear what the clinical relevance is of IL11/6 signaling in patients. In figures 1 and 2 the authors try to provide a rationale for IL11 inhibition in gastro-intestinal patients, however these data are not really convincing. IL11/IL6 related gene expression levels are only slightly enhanced in some of the subtypes IL11RA, IL6ST and IL6RA in GS for gastric cancer, IL6ST and IL6 in MSI-H for colorectal cancer. However they state: In colorectal cancer, we detected higher expression of IL11, IL6, IL6RA and IL6ST in the MSI-High and MSI-Low subsets than in the Microsatellite Stable (MSS) subtypes (Fig. 1C).

This does not correspond with the data shown in the figure, no significant differences for IL11 or IL6RA. Also MSI-L seems to be more similar to MSS than to MSI-H.

Furthermore they show differential gene expression levels for high and low risk patients. Whereas IL11 and IL6ST are both increased in high risk patients, IL11RA is lower in high risk gastric cancer patients.

In Fig. 2B/D KM plots of high and low risk patients are shown, however, I don't think this is relevant for the data shown, the authors should dichotomize the patients by high/low IL11, IL11RA, IL6ST etc and then show eventual differences in survival. It is obvious that high risk patients will have a worse survival.

I do not see how the authors conclude that "elevated expression of IL11 signaling, rather than IL6 components correlates with increased risk of recurrent disease." This does not follow from figures 1 and 2. Together with the results shown in Fig 3, where the disrupting effect of BZA on IL11 signaling is shown, which seems to be comparable to the effect on IL6 mediated growth (3B), and Fig 4, where the binding sites for IL6 and IL11 on GP130 are compared to that of BZA, (very similar), it cannot be concluded that IL11 signaling is more important than IL6. Nevertheless, it seems convincing that BZA is able to block IL11 mediated GP130 signaling, however, in principle the effects on tumor (cell) growth could still also be mediated via IL6.

Next the authors used the IL11 dependent gp130-Y757F mouse model to show that BZA is able to inhibit IL11 dependent tumor growth, independent of its SERM function. Furthermore, they show that also murine APC driven colorectal tumors can be inhibited by BZA, via STAT3 reduction, independent of APC or the immune system.

In summary, the authors show that bazedoxifene, which is an FDA approved compound, can be used as an anti-tumor treatment for CRC or gastric cancer patients, most likely due to IL11/IL6 inhibition. Since the anti-tumor effects of IL11 inhibition have been shown before, also by the authors as is stated several times in the manuscript, the novelty of this manuscript should mainly be in the repurposed use of BZA.

However, in a recent paper (Li, S., Tian, J., Zhang, H. et al. Apoptosis (2018)) this effect of BZA on colon cancer was shown already, both in vitro and in vivo, and in combination with chemotherapy. Also in other cancer types a similar mechanism has been proposed. Therefore I would not recommend this manuscript for publication in EMBO mol med.

Other comments:

It is unclear what the differences are between Fig 3 A and Supp Fig 1 A, it seems to me that they should represent a similar experiment, however in S3A with IL11Ra only, the reporter is already induced, whereas in fig 1 this is not the case.

Corroborating the selective effects of bazedoxifene on STAT3 activity (Fig. 2 and Fig S2B), tamoxifen treatment failed to suppress IL11-mediated cell proliferation of BAF/03 cells expressing human IL11Ra (Fig. S1C).
Fig 2 and S2B do not refer to STAT3 activity.

IC50 values are mentioned, however in the graphs shown, these are the highest concentrations of BZA used, are higher concentrations lethal or not working?

Figure 5:

The organoid size seems to be very small, after 6+7 days of culturing, it seems that the BZA treated cells are dying?

Why are the protein blots not shown for CRC4, or the organoid sizes of CRCs 1-3, to make a proper correlation.

It is unclear why gene expression of SPP1 is shown in 6G and 7H

Fig 6G, 7H,I: according to the legends the data is normalized to the mean value of vehicle control, this implies the black bars should be 1, but they are not.

There is no section for RNA isolation/ QPCR in the methods

Fig. 8B no AZD1480 is shown

Referee #2 (Comments on Novelty/Model System for Author):

That BZA would act via gp130-STAT3 has recently been established and some in vitro work in colon cancer published, anticipating the conclusions here; however, this is a systematic study with multiple mouse models (as appropriate as are available) along with careful structural work and reasonable dose-response analyses with molecular analysis overcomes the slight lack of novelty. Also, the potential medical relevance and impact is high.

In the current version of the manuscript, the first two figures on the TCGA analysis are very confusing, and I don't see how they do anything positive and, conversely, might discourage many readers from getting to the far more impactful later figures and analyses.

Referee #2 (Remarks for Author):

This is a dense manuscript extensively analyzing the anti-gp130 properties of the SERM BZA. Though the results would largely be predicted based on previous literature, these preclinical studies and pharmacological, molecular, histological, and structural analyses must be done. Overall, technically, the experiments are sound, well-controlled, and well-described other than a few small problems here and there. The biggest issues are that Figs. 1 and 2 based on the TCGA are highly confusing and don't seem to add anything to the manuscript. Also, much is made in the Introduction and TCGA analysis of IL-11, yet all the subsequent experiments for the most part don't distinguish between the relative contribution of IL-6 and IL-11 to BZA's anti-tumor effects.

More detailed points:

The description of olamkicept (sgp130Fc) in the Introduction is confusing. This antagonist is a dimerized gp130 which was designed to bind soluble IL-6R-IL-6 pairs to block interaction in trans with gp130 on target cells. If IL-11 (or IL-22 for that matter) also can couple with a soluble IL-11r, then this agent would work against IL-11, too. If there is no such IL-11 mechanism of action then it is not really a parallel comparison, as sgp130Fc is not supposed to have action against the direct IL-

6 interaction with membrane-bound IL-6r. The manuscript is often biased towards an IL-11 interpretation of results that is not warranted by the results, and this passage is a good example.

Fig. 1A How are the percentages calculated in Fig. 1A? Is this simply the total of each gene's gene alterations (mathematical union or intersection), such that patients with mutations in more than one of these genes get counted more than once or only once? Another way to put it: the fact that 16% have mutations in one of the genes, does that mean that 84% of people with gastric cancer have mutations in none of the genes in question?

Fig. 1B: It is really unclear what any of this analysis is supposed to mean. What is the null hypothesis here? That genetic alterations would be the same in all types of cancer or that one type would be expected to have more? My suspicion is that you could get this sort of pattern (of some genes being increased in some tumor subtypes) with just about any set of 5 genes, as you are comparing 5 separate genes in 4 separate categories, so there is a lot of chance for there to be statistical differences simply based on multiple hypothesis testing error. But, again, it is hard to know what to conclude from this analysis.

Why were IL11, IL11RA, and IL6ST the only ones chosen for analysis in Fig. 2? What do the authors make of the flip in prognostic value of IL11RA between stomach and colon cancer? What is the green-red (high-low risk) stratification based on in Fig. 2B,D? Is it in any way correlated with expression levels of the IL6/IL11 pathway genes? If so, this is not clear, and it seems as though low-high risk is generated by TCGA? Wouldn't the authors want to show that some combination of alterations (genetic or expression based) in the signaling they are interested in can be used to stratify?

Again, how the presentation in Figs 1 and 2 address the null hypothesis, wherein this cytokine signaling pathway of interest does not have any particular role in development or progression of cancer, is not clear at all.

The problem of understanding the significance of Figs. 1 and 2 is compounded by the authors saying that they find (p. 8) that IL11 signaling, rather than IL6 components correlates with increased risk of recurrent disease, yet whatever data the authors have to suggest IL6 is not important is not shown, yet the first two figures with data whose interpretation is not entirely clear is shown??

Structural analysis of potential active sites-drug interface: Could Tamoxifen be shown as a negative control in one of these views?

Fig. 6B the BZA treated stomach picture has no contrast, and anatomic boundaries are uninterpretable. A better image should be provided. 6G: IL-6 cytokine must certainly be significantly increased (though it is not analyzed statistically apparently). What would be the significance? Feedback inhibition?

Fig. 7I: what the normalizer here is unclear (ie what is relative expression of 1?). The legend says relative to vehicle, but in SOX9, eg, both VEH and BZA are less than 1. Are they reduced in tumors? Is this vehicle control a non-Tamoxifen-treated vehicle control?

All the cells in the representative photomicrographs in SF6 are pSTAT3 in both VEH and BZA, so it's not clear where the quantification plot at right comes from.

STATS: many of the analyses involve more than two groups for which T-test is not a valid method of estimating statistical significance.

A recent paper in Apoptosis shows also BZA via GP130 anti-tumor effects in colon cancer cell lines.

Author appeal and editor reply

13 September 2018

Thank you for your email and feedback on our manuscript. We are very appreciative of the thoughtful critique and that the referees were positive about our findings on repurposing bazedoxifene for gastrointestinal cancers. We also acknowledge the concerns you have regarding the novelty of our findings. However, the conflicting opinions of the two reviewers on this matter has

prompted my response to the decision made. I would be really grateful if you were to find time to evaluate our response on this matter.

Regarding novelty of our findings, we are in the agreement with the opening statement of reviewer # 2; (quoted below):

“That BZA would act via gp130-STAT3 has recently been established and some in vitro work in colon cancer published, anticipating the conclusions here; however, this is a systematic study with multiple mouse models(as appropriate as are available) along with careful structural work and reasonable dose-response analyses with molecular analysis overcomes the slight lack of novelty. Also, the potential medical relevance and impact is high. “

The quality of data presented in previous publication (in the journal Apoptosis), which assesses bazedoxifene in a colon cancer xenograft lacks depth, in comparison to our study. Some points of novelty about our study are listed here and relate to the (1) use of our specialized model systems and the demonstration of IL11 signalling inhibition as a mechanism by which bazedoxifene reduced tumour burden in models employed. This has not been shown before.

Novelty of the animal models and the demonstrated of the specific blockade of IL11-dependent cancer in vivo include the use of the:

1. gp130FF gastric tumour model which is reliant on IL11 for tumour development and growth. The current study is the first proof-of-principle in vivo that bazedoxifene is the first small molecule inhibitor that specifically targets an IL11-dependent tumor development pathway. The publication (in Apotosis) highlighted by reviewers uses a human colon cancer xenograft, and is distinctly lacks experiments to provide mechanistic proof of the selective inhibition of IL11 or IL6 in their chose model system.
2. Epithelial-specific APC mutation model of colon cancer to demonstrate that in an immune-competent setting, the decrease in tumour growth with bazedoxifene treatment is dependent on processes of proliferation and apoptosis and not immune cells.
3. Cytokine-dependent cell proliferation assays to demonstrate specificity of the drug receptor interactions and the first demonstration of a mechanism of action of bazedoxifene on gp130-receptor conformations with IL6R, IL11R, LIFR and Lgp130.

The seminal work of my mentor and co-senior author on this manuscript, Prof Ernst (included in this email) demonstrated the impact of IL11 as the key cytokine in driving cancers in the gastric and colon models employed in our studies (Putoczki et al 2013, Cancer Cell). Hence our conclusions on the effects of bazedoxifene on IL11 suppression are well researched and performed in the most relevant models available. Hence I feel that our findings contribute to new mechanistic knowledge that is valuable for the translational of such a finding into the clinic as a new treatment opportunity. The clinical relevance of repurposing bazedoxifene will appeal a broad readership.

I sincerely hope that EMBO Molecular Medicine and the reviewers of this manuscript will consider my proposal of resubmitting a corrected draft for reconsideration.

If appropriate, I am also happy to speak with you over the phone to provide more information pertaining to our study, the approaches used and the validity of our research findings.

EDITOR REPLY

Please accept my sincere apologies for only getting back to you today. We have been extremely busy with newly submitted articles and staff travelling to conferences, reducing our capacity and availability for dealing with appeals.

I have obtained extra comments from both referees, included below:

#ref2

"I think the authors really shot themselves in the foot with figures 1 and 2 which try to tie in with their (presumably, as the authors were anonymized) work on IL-11, etc. But the analysis was poorly explained, executed, and analyzed and led to little justification for the work to follow. In fact, it led to far more confusion than if they'd just started with a paper to analyze BZA in gastric cancer. Most

of the highly negative comments from myself and others all have to do with figures 1 and 2 and how that was supposed to rationalize what follows.

I do not have strong feelings about relative novelty. I thought, in the end, preclinical work will need to be done, and the preclinical work on BZA in the stomach was reasonably solid. It is the first such work in the stomach, though, as pointed out, this has been done before in other organs. That may be reason to kill it for EMBOMolMed [...]. Thus, the work is important from a translation, application point of view."

#ref1

"I largely agree with ref2 about figures 1 and 2, which in my opinion do greatly reduce the confidence in the rest of the paper. Indeed, the other figures, apart from my previous comments, are more or less solid. However, since the effects of BZA have already been shown in animal models of colorectal cancer, I still think the presented work does not progress the translational knowledge enough to justify publication"

We have discussed these comments and initial decision within the team and came to the conclusion that, if you can convincingly fix the issues shared by both the reviewers regarding figures 1 and 2 and address the rest of the critics to satisfy referees, we would encourage resubmission of your study within 3 months.

I thank you for your patience and interest in EMBO Molecular Medicine.

Resubmission - authors' response

28 November 2018

Response to **Reviewer #1**

Q1. *It is unclear what the differences are between Fig 3 A and Supp Fig 1 A, it seems to me that they should represent a similar experiment, however in S3A with IL11Ra only, the reporter is already induced, whereas in fig 1 this is not the case.*

Response: The reviewer is correct that these panels depicted a repeat of the same observations; we have now removed the graph from Supp Fig 1 A.

Q2. *“Corroborating the selective effects of bazedoxifene on STAT3 activity (Fig. 2 and Fig S2B), tamoxifen treatment failed to suppress IL11-mediated cell proliferation of BAF/03 cells expressing human IL11Ra (Fig. S1C)”. Fig 2 and S2B do not refer to STAT3 activity.*

Response: As requested by the reviewer, the figure reference has been corrected.

Q3. *IC50 values are mentioned, however in the graphs shown, these are the highest concentrations of BZA used, are higher concentrations lethal or not working?*

Response: Higher concentrations do show cell death in some of the assay systems tested. However, we would like to point out that we have not referred to specific IC50 values in our manuscript.

Q4. *Figure 5: The organoid size seems to be very small, after 6+7 days of culturing, it seems that the BZA treated cells are dying?*

Response: We have developed protocols that allow monitoring of both growth/proliferation of emerging organoids (during the first 5 days) and survival of organoids (days 5-7). As the reviewer points out correctly, we observed the major effect of BZA to reduce survival of cells in the organoids, consistent with our observation of BZA-treatment of tumor bearing mice induces reduces expression of survival genes and promotes apoptosis.

Q5. *Why are the protein blots not shown for CRC4, or the organoid sizes of CRCs 1-3, to make a proper correlation?*

Response: The organoid (CRC4) experiment and the CRC 1-3 experiments were conducted by different collaborators and the amount of organoid protein we can recover from the

matrigelembedded organoids is insufficient to for reliable assessment of pSTAT3 by Western blots analysis.

In light of this, we were careful as to not over-state in the manuscript a causal relationship between BZA treatment, pSTAT3 and organoid size in the organoid assay, although this strongly implied from the CRC 1-3 experiments carried out in conventional tissue cultures.

Q6. *It is unclear why gene expression of SPP1 is shown in 6G and 7H*

Response: SPP1 was identified as an IL11-regulated gene in the antrum tumors treated with IL11-Mutein (Putoczki *et al* Cancer Cell 2013). Hence it was included in the assessment of selected target genes. However, as the function of SPP1 is associated with bone extracellular matrix activity rather than a cancer cell intrinsic activity relevant in the context of the manuscript, we have removed SPP1 mRNA transcript data from the figure in the amended manuscript for clarity.

Q7. *Fig 6G, 7H,I: according to the legends the data is normalized to the mean value of vehicle control, this implies the black bars should be 1, but they are not. There is no section for RNA isolation/ QPCR in the methods.*

Response: The reviewer is correct, and we have corrected this for the figures mentioned. We have updated Materials and Methods section.

Q8. *Fig. 8B no AZD1480 is shown.*

Response: Thank you for highlighting, this has been corrected in the results section to match the figure content.

Response to **Reviewer #2**

Q1. *“...much is made in the Introduction and TCGA analysis of IL-11, yet all the subsequent experiments for the most part don't distinguish between the relative contribution of IL-6 and IL-11 to BZA's anti-tumor effects”.*

Response: We would like to point out to the reviewer that we have deliberately chosen cancer models where we have previously shown a more prominent effect of IL11 over IL6 (Putoczki *et al* Cancer Cell, 2013). We felt this to be important to substantiate our claim that BZA can suppress tumor growth by suppressing IL11. However, given our structural analysis and the existing data for BZA on IL6, we cannot assess the contribution of IL6-(trans signaling) mediated promotion of tumor growth and how suppression of this activity by BZA contributes to the overall effect conferred by BZA-treatment of tumor-bearing mice in the *Cdx2CreERT2;Apcflox* and *Lgr5CreERT2;Apcflox* models. By contrast, our genetic evidence obtained in *gp130Y757F* mice suggests quite clearly that IL6 signaling (conventional and trans-signaling) is completely dispensable and hence in this model the effect of BZA has to be mediated by inhibition of IL11 signaling (Ernst *et al.*, J Clin Invest 2008) We clearly state both of this points within the Results and Discussion section of the manuscript.

Q2. *The description of olamkicept (sgp130Fc) in the Introduction is confusing...The manuscript is often biased towards an IL-11 interpretation of results that is not warranted by the results, and this passage is a good example.*

Response: The reference to olamkicept has been removed. As for the reasons outlined to Query #1 above, we believe that an interpretation of the data with a focus on IL11 is justified.

Q3. *Structural analysis of potential active sites-drug interface: Could Tamoxifen be shown as a negative control in one of these views?*

Response: As requested by Reviewer, we have now added a panel to Supplementary Figure 2 to show *tamoxifen* as a negative control and to illustrate the structural differences between *tamoxifen* and *bazedoxifene*. Accordingly, information pertaining to these changes have been included in the Methods and Results sections.

Q4. *Fig. 6B picture contrast, and anatomic boundaries are uninterpretable. 6G: IL-6 cytokine must*

certainly be significantly increased (though it is not analyzed statistically apparently). What would be the significance? Feedback inhibition?

Response: as per the reviewer's request the contrast of that Figure has been improved. Because, the "increase" in IL6 is not statistically significant, we do not believe that a discussion on potential feedback inhibition is warranted.

Q4. *Fig. 7I: what the normalizer here is unclear (i.e. what is relative expression of 1?). The legend says relative to vehicle, but in SOX9, e.g. both VEH and BZA are less than 1. Are they reduced in tumors? Is this vehicle control a non-Tamoxifen-treated vehicle control?*

Response: We would like to clarify that the abundance of SOX9 transcripts is very low in the colon, and the data was reflective of its relative abundance to the housekeeping gene. This data has been normalised using the delta delta CT method. The control is a tamoxifen-treated vehicle control. Tamoxifen is structurally very different and is administered for 3 days at the start of the experiment. The BZA treatment is initiated 2 weeks after tamoxifen administration.

Q5. *All the cells in the representative photomicrographs in SF6 are pSTAT3 in both VEH and BZA, so it's not clear where the quantification plot at right comes from.*

Response: The quantification plot is the quantification of pSTAT3 stained cells. The Aperio software used for quantification categorizes the intensity of immunostainings into 0 (absent), 1+(weak) to 3+ (most intense stain). The graphs show staining of 2+ and 3+ scoring cells. The figure legends and Methods section have been changed accordingly.

Q6. *STATS: many of the analyses involve more than two groups for which T-test is not a valid method of estimating statistical significance.*

Response: As per the reviewer's request ANOVA and multiple comparisons with Dunnett's or Tukey tests have now been performed where appropriate and these changes are reflected in the revised text.

2nd Editorial Decision

21 December 2018

Thank you for re-submitting a revised manuscript to EMBO Molecular Medicine. We have now received the enclosed reports from the one referee that we asked to re-assess it. As you will see the reviewer is now globally supportive and I am pleased to inform you that we will be able to accept your manuscript pending the following final amendments:

1) Please address the comments of referee 1. Provide a point-by-point letter INCLUDING my comments and the reviewer's reports and your detailed responses to their comments (as Word file).

Please submit your revised manuscript within one month. I look forward to seeing a revised form of your manuscript as soon as possible.

I look forward to reading a new revised version of your manuscript as soon as possible.

***** Reviewer's comments *****

Referee #1 (Remarks for Author):

Revision of "Repurposing the selective estrogen receptor modulator bazedoxifene to suppress gastrointestinal cancer growth", Thilakasiri et al.

In this revision, the authors have improved the manuscript largely by removing the previous figures 1 and 2. Also, the story is much more consistent, starting by demonstrating the specific effect of Bazedoxifene (BZA) on IL6/11 signaling in fig 1, followed by detailed putative binding

mechanisms in fig 2. Next they show the inhibitory effect of BZA on IL11/IL6 induced STAT3 activation in cell lines and organoid cultures, as well as an IL11 dependent growth inhibitory effect on organoids.

Further, BZA treatment in vivo in 3 different relevant gastrointestinal mouse models resulted all in decreased tumor growth. This was shown to be the result of IL11/STAT3 inhibition, and independent of immune responses or alterations in Wnt signaling.

Although the novelty of the research has not really altered compared to the initial submission, the manuscript now gives a clear and thorough study of the treatment of GI cancers using BZA, covering the (molecular) mechanism, in vitro effects and convincing in vivo studies and provides strong arguments for further clinical research in this direction. I would now recommend this article for publication, when some minor changes have been addressed:

- Fig 1A: it should be indicated in the graph which compound is used
- Fig 4G: As the other reviewer previously already indicated, the IL6 expression after BZA treatment suggests an feedback mechanism, although the authors state that there is no significant difference, the size of the error bars suggest differently. However this might be the result of the way the data is showed, error bars represent SEM, this might be solved by plotting SD.
- Fig 5A: the schematic has a different timing than indicated in the legends (BZA treatment starts 7 days after TAM) and in the text (BZA treatment starts 2 weeks after TAM)
- Fig 5E: this subfigure could better be placed after 5G
- Fig 5H/I: the normalization of the VEH controls is still not set to 1 in 5H, whereas it is in 5I.
- For the flow of the story, it would be beneficial to switch the paragraph about the immune response (Fig S7/8) with the Wnt signaling paragraph (fig 6/S6) in the text.
- Fig S9 is not mentioned in the text

2nd Revision - authors' response

19 January 2019

Responses to **Referee #1**

Fig 1A. The names of the compounds used in experiments have been included in Figure 1A.

Fig 4G. P values have been included in the figure to clearly identify whether significant differences are observed.

Fig 5A. The details in the accompanying figure legend has been modified to describe that treatment of *bazedoxifene* was initiated 2 weeks after the administration of *tamoxifen* (Page 30). This corresponds now with the results text on page 10.

Fig 5E. Has been moved to after 5G as suggested and relabeled accordingly in the text (page 10) and figure legends (page 31).

Figs S7/8 and Fig 6/S6. The paragraph describing effects to tumor infiltrating immune cells has been placed after the paragraph on effects on Wnt signaling as suggested in order to improve the flow of the results section (pages 11 and 12).

Fig S9. The figure reference has been corrected in the manuscript text.

Corresponding Author Name: Ashwini Chand
Journal Submitted to: EMBO Molecular Medicine
Manuscript Number: EMM-2018-09539